# Parvifloron D from *Plectranthus*
*strigosus*: Cytotoxicity Screening of *Plectranthus* spp. Extracts

**DOI:** 10.3390/biom9100616

**Published:** 2019-10-17

**Authors:** Catarina Garcia, Epole Ntungwe, Ana Rebelo, Cláudia Bessa, Tijana Stankovic, Jelena Dinic, Ana Díaz-Lanza, Catarina P. Reis, Amílcar Roberto, Paula Pereira, Maria-João Cebola, Lucília Saraiva, Milica Pesic, Noélia Duarte, Patrícia Rijo

**Affiliations:** 1Research Center for Biosciences & Health Technologies (CBIOS), Universidade Lusófona de Humanidades e Tecnologias, 1749-024 Lisboa, Portugal; catarina.g.garcia@gmail.com (C.G.); ntungweepolengolle@yahoo.com (E.N.); ana.rebelo1490@gmail.com (A.R.); amilcar.roberto@ulusofona.pt (A.R.); m.joao.cebola@gmail.com (M.-J.C.); 2Department of Biomedical Sciences, Faculty of Pharmacy, University of Alcalá, Campus Universitario, 28871 Alcalá de Henares, Spain; ana.diaz@uah.es; 3LAQV/REQUIMTE, Laboratório de Microbiologia, Departamento de Ciências Biológicas, Faculdade de Farmácia, Universidade do Porto, Rua de Jorge Viterbo Ferreira n. 228, 4050-313 Porto, Portugal; claudiasantos26@gmail.com (C.B.); lucilia.saraiva@ff.up.pt (L.S.); 4Institute for Biological Research, “Siniša Stanković”, University of Belgrade, Despota Stefana 142, 11060 Belgrade, Serbia; tijana.andjelkovic@ibiss.bg.ac.rs (T.S.); jelena.dinic@ibiss.bg.ac.rs (J.D.); camala@ibiss.bg.ac.rs (M.P.); 5Instituto de Investigação do Medicamento (iMed.ULisboa), Faculdade de Farmácia, Universidade de Lisboa, 1649-003 Lisboa, Portugal; catarinareis@ff.ulisboa.pt; 6CERENA–Centre for Natural Resources and the Environment, Instituto Superior Técnico (IST), Universidade de Lisboa, Av. Rovisco Pais, 1049-001 Lisbon, Portugal

**Keywords:** *Plectranthus*, *P. strigosus*, antimicrobial, cytotoxicity, parvifloron D

## Abstract

The *Plectranthus* genus is commonly used in traditional medicine due to its potential to treat several illnesses, including bacterial infections and cancer. As such, aiming to screen the antibacterial and cytotoxic activities of extracts, sixteen selected *Plectranthus* species with medicinal potential were studied. In total, 31 extracts obtained from 16 *Plectranthus* spp. were tested for their antibacterial and anticancer properties. Well diffusion method was used for preliminary antibacterial screening. The minimum inhibitory concentration (MIC) and minimal bactericidal concentration (MBC) values of the five most active acetonic extracts (*P. aliciae*, *P. japonicus*, *P. madagascariensis var. “Lynne”*, *P. stylesii*, and *P. strigosus*) were determined. After preliminary toxicity evaluation on *Artemia salina* L., their cytotoxic properties were assessed on three human cancer cell lines (HCT116, MCF-7, and H460). These were also selected for mechanism of resistance studies (on NCI-H460/R and DLD1-TxR cells). An identified compound—parvifloron D—was tested in a pair of sensitive and MDR-Multidrug resistance cancer cells (NCI-H460 and NCI-H460/R) and in normal bronchial fibroblasts MRC-5. The chemical composition of the most active extract was studied through high performance liquid chromatography with a diode array detector (HPLC-DAD/UV) and liquid chromatography–mass spectrometry (LC–MS). Overall, *P. strigosus* acetonic extract showed the strongest antimicrobial and cytotoxic potential that could be explained by the presence of parvifloron D, a highly cytotoxic diterpene. This study provides valuable information on the use of the *Plectranthus* genus as a source of bioactive compounds, namely *P. strigosus* with the potential active ingredient the parvifloron D.

## 1. Introduction

The *Plectranthus* genus (Lamiaceae) comprises nearly 350 different species with a vast distribution and is recognized for its rich bio- and ethnobotanical diversity [1,2,3] (see Appendix A). Its antibacterial, antifungal, antiviral, and antiproliferative potential have been well-described in literature [2]. These characteristics contribute to its popular use in traditional medicine, mainly through herbal preparations, thus having major importance in the provision of primary health care [3,4,5].

Cancer is still one of the major leading causes of death worldwide and the success of anticancer therapies is dependent on the development of multidrug resistance (MDR) [5,6]. Plants have long been used for the treatment of cancer and there are notable anticancer plant-derived drugs used in clinical practice, namely vinca alkaloids (vinblastine and vincristine) and paclitaxel [7,8]. In addition, several plant-derived products, namely terpenoids, and extracts have been described for their capability to inhibit P-glycoprotein (P-gp), and thus modulate MDR [9].

Terpenoids are also considered as responsible for the antimicrobial properties attributed to the aforementioned genus [3]. The increasing resistance to antibiotics and rising range of infections have become a public health-concerning issue [10].

Diterpenes are frequently found in the *Plectranthus* genus and are compounds of high interest, considering their wide spectrum of biological activity, characterized by antimicrobial and anti-inflamatory activity, but also, anticancer properties [11]. The majority of diterpene metabolites isolated from *Plectranthus* spp. belong to abietane, kaurane, and labdane classes [11]. The more prominent functional groups are phenol or quinone diterpenes belonging to royleanone, coleon, or parvifloron abietanes, which have been shown to be potent anticancer agent classes [12,13]. It is known that the major component of the essential oil of *P. madagascariensis* is the diterpene 6,7 dehydroroyleanone, which is easily identified due to its characteristic orange reddish color [14,15] and has already shown cytotoxic properties [16,17]. On the other hand, it is also ascribed antitumoral activity by a number of other *Plectranthus* species such as *P. amboinicus*, *P. barbatus*, and *P. hadiensis* [2], so the screening of their extracts is a significant achievement. Other previous studies showed that natural compounds with antitumor activity, isolated from *Plectranthus*, induce apoptosis by caspases activation. The discovery of natural tumor-promoting phorbol esters acting as protein kinase C (PKC) activators led to a new interest in the role of these proteins, but also, and more importantly, to new studies focusing on diterpenes as PKC modulators. It is known that some of the PKC isoforms are implied in tumoral regression, while others are involved in tumoral invasion, so modulating its activity is an important issue [18]. Having this in mind, the present study was aimed at screening the antimicrobial and cytotoxic properties of several *Plectranthus* spp. extracts. These goals were accomplished by comparing the efficacy of 31 plant extracts and identifying the bioactive component responsible for the extracts activity.

## 2. Experimental

### 2.1. Plant Material

*Plectranthus* spp. (*Lamiaceae* family) were cultivated in Instituto Superior de Agronomia Campus (Lisbon University), from seeds provided by the herbarium of the Botanical Garden of Lisbon, Portugal. The air dried and powdered whole plants were selected based on their ethnopharmacological features: *Plectranthus aliciae* (Codd) van Jaarsv. and Edwards, *Plectranthus amboinicus* (Lour.) Spreng., *Plectranthus barbatus* Andrews, *P. ecklonii* Benth., *P. fruticosus* L’Hér., *Plectranthus hadiensis* (Forssk.) Schweinf. ex Spreng. var. *hadiensis, P. hereroensis* Engl., *Plectranthus japonicus* Koidz., *Plectranthus madagascariensis* (Pers.) Benth. var. *madagascariensis*, *P. madagascariensis* (Pers.) Benth var. ‘Lynne’, *Plectranthus malvinus* van Jaarsv. and T.J. Edwards, *Plectranthus oertendahlii* Th. Fr., *Plectranthus reflexus* van Jaarsv. and T.J. Edwards, *Plectranthus stylesii* Edwards, *P. strigosus* Benth. ex E.Mey., and *Plectranthus zuluensis* T. Cooke. The plant names were verified with the Index Kewensis [19]. The plants were grown in Parque Botânico da Tapada da Ajuda from cuttings provided through the Kirstenbosch National Botanical Gardens, South Africa; were collected between 2007 and 2008, always in June and September; and voucher specimens were deposited in the Herbarium “João de Carvalho e Vasconcellos” of the “Instituto Superior de Agronomia”, Lisboa (LISI), Portugal [20].

Traditional uses of the mentioned plants and their voucher numbers are displayed in Appendix A.

### 2.2. Extraction Procedure of Plectranthus spp.

Thirty-one extracts were prepared from sixteen different *Plectranthus* species. The air-dried leaves of each plant were milled in a kitchen mill and weighed. The extraction procedure was carried out in triplicate in an ultrasonic (US) bath (Bandelin SONOREX RK 510H, Berlin, Germany) for 60 min at 25 °C—using acetone, ethanol, and/or methanol—in a concentration of 10% (*w*/*v*). The suspensions were then filtered using Filtrer Laurent—Prat demas 150 mm (Couze-St-Front, France). After each filtration, the solvent was removed in a rotary evaporator (Sigma-Aldrich, IKA HBR 4 basic heating bath, Essen, Germany) at 40 °C. After submitting the suspensions to ultrasounds three times, the final extracts were weighed, and the yield of extraction determined. The percentage yield for each extract was determined (Appendix A).

### 2.3. Microorganisms Used and Growth Conditions

Bacterial strains *Enterococcus faecalis* (ATCC 29212), *Escherichia coli* (ATCC 25922), *Pseudomonas aeruginosa* (ATCC 27853), and *Staphylococcus aureus* (ATCC 25923), and yeast strains *Candida albicans* (ATCC 10231) and *Saccharomyces cerevisiae* (ATCC 9763) were chosen based on their clinical and pharmacological importance. In addition, *S. cerevisiae* was added due to its general toxicity screening value. The tested bacteria were precultivated on Mueller Hinton agar and yeasts in Sabouraud agar (Biokar diagnostics, Allonne, France). Cultures of microorganisms were grown at 37 °C (30 °C for the yeasts) for 24 h and were maintained on respective agar slants at 4 °C.

### 2.4. Antimicrobial Screening Assays

#### 2.4.1. Well Diffusion Method

The extracts were reconstituted in DMSO—Dimethyl Sulfoxide—to a 1 mg/mL concentration as well as the stock solutions of reference antibiotics (vancomycin and norfloxacin, Gram-positive and -negative, respectively). Nystatin was used as the positive control for the yeasts. Afterwards, Petri dishes containing 20 mL of solid Mueller–Hinton culture medium (Sabouraud for the yeasts) were inoculated with 100 μL of microorganism suspension (correspondent to a 0.5 McFarland standard solution). The inocula were evenly harvested on the medium surface using a sterile swab [20]. Wells of approximately 5 mm in diameter were punched aseptically in the medium, with a sterile glass Pasteur pipette and 50 μL of each extract and antimicrobial agent were introduced into the well [20,21]. The solvent DMSO was used as negative control. Plates were then incubated at 37 °C (30 °C for the yeasts) for 24 h. The antimicrobial activity was then evaluated by measuring the diameter (mm) of the inhibition zone formed around the wells and the results compared to positive and negative controls [20]. This method was tested at least in triplicate.

#### 2.4.2. Determination of Minimum Inhibitory Concentration

Only extracts that showed inhibitory activity on the well diffusion method were tested for this method, so only the bacteria were active. After screening the antimicrobial activity of the plant extracts, the microplate broth microdilution method was performed for those that were considered active (inhibition zone > 10 mm). As such, 100 μL of liquid Mueller–Hilton medium was distributed in each well of a 96-well plate. The first well of each row contained 100 μL of extract, the positive control, or negative control solutions (1 mg/mL concentration). Serial dilutions were made to 1:2 proportion between each row of wells (1.95–500 μg/mL range). Finally, 10 μL of bacterial suspension of 0.5 McFarland density were added to each well. Plates were then incubated at 37 °C for 24 h, allowing the bacterial growth. The latter was measured with an absorbance microplate reader (Thermo Scientific Multiskan FC, Loughborough, UK) set to 620 nm. All assays were carried out in triplicate for each tested microorganism [20].

#### 2.4.3. Determination of Minimal Bactericidal Concentration

After concluding the minimum inhibitory concentration (MIC) assay, the minimal bactericidal concentration (MBC) assay was done accordingly to the agar dilution method. Thus, a swab from each well where no growth was observed—containing the tested extract concentration series—was plated on a Mueller Hinton Agar [21,22,23]. The Petri plates were then incubated for 24 h at 37 °C, for further evaluation. The lowest concentration that revealed no visible bacterial growth after subculturing was taken as its MBC. Only the bacteria showed inhibition growth.

### 2.5. Cell Lines and Reagents

#### 2.5.1. Chemicals

MTT (3-(4,5-dimethyl-2-thiazolyl)-2, 5-diphenyl-2H-tetrazolium bromide), rhodamine 123 (Rho123), and dimethylsulfoxide (DMSO) were acquired from Sigma-Aldrich Chemie GmbH. Fetal bovine serum (FBS), RPMI 1640 medium (Roswell Park Memorial Institut), DMEM (Dulbecco’s Modified Eagle Medium), penicillin–streptomycin solution, antibiotic–antimycotic solution, l-glutamine, and trypsin/EDTA were purchased from PAA (Vienna, Austria).

#### 2.5.2. Cell Culture Maintenance

Human colon (HCT116) and breast (MCF-7) adenocarcinoma, and non-small cell lung carcinoma (NCI-H460) cell lines were purchase from ATCC. All cancer cells were cultured in RPMI-1640 medium with ultraglutamine (Lonza, VWR, Carnaxide, Portugal), and supplemented with 10% fetal bovine serum (FBS; Gibco, Alfagene, Carcavelos, Portugal). Cells were maintained at 37 °C in a humidified atmosphere of 5% CO_2_.

Sensitive non-small cell lung carcinoma NCI-H460, colorectal carcinoma DLD1 cell lines, and normal human embryonal bronchial fibroblasts MRC-5 were purchased from American Type Culture Collection (Rockville, MD, USA).

NCI-H460 and DLD1 cells were maintained in RPMI 1640 medium containing 10% heat in activated FBS, 2 mM l-glutamine, 4.5 g/L glucose, 10,000 U/mL penicillin, 10 mg/mL streptomycin, and 25 mg/mL amphotericin B solution at 37 °C in a humidified 5% CO2 atmosphere. NCI-H460/R cells were originally selected from NCI-H460 cells and cultured in a medium containing 100 nM doxorubicin (DOX) [23]. DLD1-TxR cells were selected by continuous exposure to stepwise increasing concentrations of paclitaxel (PTX) from DLD1 cells [24].

MRC-5 cells were cultured in DMEM supplemented with 10% FBS, 4 g/L glucose, l-glutamine (2 mM), and 5000 U/mL penicillin, 5 mg/mL streptomycin solution at 37 °C in a humidified 5% CO_2_ atmosphere.

All cell lines were subcultured at 72 h intervals using 0.25% trypsin/EDTA and seeded into a fresh medium at the density of 8000 cells/cm^2^.

### 2.6. Preliminary Evaluation of General Toxicity on Artemia salina Model

Aiming at the evaluation of the preliminary toxicity of the selected *Plectranthus* spp. extracts, a lethality test on *Artemia salina* L. was carried out [25]. The general toxicity on *A. salina* was tested according to the method of Zhang et al. 2012 [26], with slight alterations.

Brine shrimp cysts (obtained from JBL GmbH and Co. KG, D-67141 Neuhofen Germany) were hatched in artificial sea water with a concentration of 35 g/L. A container with two connected chambers with a simple handmade communicating vessel was used for brine shrimp hatching. An air pump inserted in the container ensured a regular air flow supply, thus saturating the artificial sea water for successful hatching. The cysts were then incubated for 48 h at 24 °C. Each extract was tested at a concentration of 100 ppm. To do so, ten to fifteen nauplii were transferred into wells of 24-well plates containing artificial sea water, and 100 μL of each extract was added to the wells (final volume per well: 1 mL). After 24 h exposure to the extracts (24 °C), the number of dead nauplii (mortality rate (%)) was determined (Equation (1)). Also, LC_50_ (Lethal Concentration, 50%) values (μg/mL) were calculated for the most toxic extracts. DMSO was used as the solvent and was kept at 10% (*v*/*v*) in all samples tested.
(1)$$Mortality\;rate\;\left(\%\right)=\frac{Total\;nauplii-Alive\;nauplii}{Total\;nauplii}$$

### 2.7. Cytotoxicity Screening

#### Sulforhodamine B Assay for the Assessment of Extracts Effects

The effect of plant extracts on the in vitro growth of human tumor cell lines was evaluated using the sulforhodamine B (SRB) assay, as described [27,28]. Briefly, cells were seeded in 96-well plates, at a final density of 5.0 × 10^3^ cells/well, and incubated for 24 h. Cells were then incubated with serial dilutions of plant extracts (from 1.56 to 50 μg/mL) for 48 h. The solvent (DMSO) was used as negative control at the same concentration as the maximum tested in the assay. The concentration of compound that causes a 50% reduction in the net protein increase in cells during treatment (GI_50_, growth inhibition of 50%) was subsequently calculated.

NCI-H460, NCI-H460/R, DLD1, and DLD1-TxR cells grown in 25 cm^2^ tissue flasks were trypsinized, seeded into flat-bottomed 96-well tissue culture plates (2 × 10^4^ cells/well), and incubated overnight. Treatments with extracts (5–100 μg/mL) lasted 72 h. The cellular proteins were stained with SRB, following slightly modified protocol of Skehan et al. [28]. Briefly, the cells in 96-well plates were fixed in 50% trichloracetic acid (50 μL/well) for 1 h at 4 °C, rinsed in tap water, and stained with 0.4% (*w*/*v*).

SRB in 1% acetic acid (50 μL/well) for 30 min at room temperature. The cells were then rinsed three times in 1% acetic acid to remove the unbound stain. The protein-bound stain was extracted with 200 μL 10 mM Tris base (pH 10.5) per well. The optical density was read at 540 nm with correction at 670 nm in a LKB 5060-006 Micro plate Reader (Vienna, Austria). The concentration of each drug that inhibited cell growth by 50%. (IC_50_) was calculated by nonlinear regression analysis using log (inhibitor) vs. normalized response in GraphPad Prism 6 software. Values represent means ± SE (Standard Error) from at least three independent experiments, each performed in triplicate.

### 2.8. Liquid Chromatography–Mass Spectrometry Analysis

Liquid chromatography–mass spectrometry (LC–MS)/MS analyses were performed on a Waters Alliance 2695 high performance liquid chromatography (HPLC) separation module connected to a Micromass Quattro Micro triple Quadropole (Waters, Dublin, Ireland) Mass Spectrometer equipped with a Waters electrospray ionization (ESI) source. Mass spectrometer was operated in positive mode and spectra were recorded in the *m*/*z* 60 to 800 Da range. In order to optimize detection conditions, a standard solution of parvifloron D was infused in the mass spectrometer and capillary and cone voltages, and collision energy were tuned to maximize (recursor ion > product ion) transition signal. ESI capillary voltage was optimized to 3 kV, cone voltage was set on 30 V. Source and desolvation temperature were adjusted to 120 °C and 350 °C, respectively. High purity nitrogen (N_2_) was used as drying gas and as a nebulizing gas. Ultra-high purity argon (Ar) was used as collision gas. A reversed phase WatersTM Atlantis C-18 (Waters, Milford, MA, USA; 5 μm × 2.1 mm × 150 mm) was used, with an injection volume of 15 μL. The mobile phase consisted in 0.5% formic acid in Milli-Q water (eluent A) and acetonitrile (eluent B). A flow rate of 0.2 mL/min was used, with the following gradient program: 0–15 min from 60% to 20% A, 15–20 min at 20% A, 20–30 min 60% A. MassLynx software (Waters, version 4.1) was used for data acquisition and processing.

The quantification of the identified compound was carried out in an Agilent Technologies 1200 Infinity Series system with diode array detector (DAD; Agilent, Santa Clara, CA, USA), equipped with a Zorbax XDB-C18, (250 × 4.0 mm i.d., 5μm) column, from Merck and ChemStation Software (Hewlett-Packard, Alto Palo, CA, USA). Each extract was analyzed (after 20 μL injection) and a gradient elution mixture composed of solution A (methanol), solution B (acetonitrile), and solution C (0.3% trichloroacetic acid in water) was used as follows: 0 min, 15% A, 5% B, and 80% C; 20 min, 70% A, 30% B, and 0% C; 25 min, 70% A, 30% B, and 0% C; and 28 min, 15% A, 5% B, and 80% C. The flow rate was set at 1 mL.min^−1^. For quantification and identification purposes, the compounds present in the samples were compared with calibration standards. Compound identification was based on ultraviolet (UV) spectra comparison and retention time. The time of analysis was of 28 min, including the stabilization of the RP-18 column. All analyses were performed in triplicate.

### 2.9. Anticancer Effects of the Identified Compound

The anticancer properties of the parvifloron D present in the most active extract were assessed through the use of a model system of NCI-H460 and its resistant counterpart NCI-H460/R, along with normal human embryonal bronchial fibroblasts MRC-5. MTT test was applied after 72 h incubation with diterpenes, applied within a range of concentrations 2.5–50 μM.

#### 2.9.1. Cell Viability Assessed by MTT Assay

MTT assay is based on the reduction of 3-(4, 5-dimethyl-2-thizolyl)-2,5-diphenyl-2H-tetrazolium bromide into formazan dye by active mitochondria of living cells. Briefly, cells grown in 25 cm^2^ tissue flasks were trypsinized, seeded in 96-well tissue culture plates (2,000 cells/well), and incubated overnight in 100 mL of appropriate medium. After 24 h, cells were treated with increasing concentrations of parvifloron D (2.5–50 μM). All treatments lasted 72 h. At the end of treatment period, 100 mL of MTT solution (1 mg/mL) was added to each well and plates were incubated at 37 °C for 4 h. Formazan product was dissolved in 200 mL DMSO. The absorbance of obtained dye was measured at 540 nm using an automatic microplate reader (LKB 5060-006 Micro Plate Reader, Vienna, Austria).

Relative cell growth rate (CG) was determined according to the following equitation:

CG (%) = (A treated sample/A untreated control) × 100,
(2)
where A is absorbance. IC_50_ value was defined as concentration of each drug that inhibited cell growth by 50%. IC_50_ was calculated by nonlinear regression analysis using log (inhibitor) vs. normalized response in GraphPad Prism 6 software.

#### 2.9.2. Assay for P-gp Inhibiting Activity

The function of P-gp was analyzed by flow-cytometry utilizing the ability of its substrate Rho123 to emit fluorescence. Studies were carried out with parvifloron D. Dex-VER was used as a positive control. The MDR cells (NCI-H460/R) were grown to 80% confluence in 75 cm^2^ flasks, trypsinized, and resuspended in 5 mL centrifuge tubes in a Rho123-containing medium (5 μM Rho123). The cells were treated with Dex-VER and parvifloron D (5 μM) and incubated at 37 °C in 5% CO_2_ for 30 mmL of cold phosphate-buffered saline, and analyzed usinin. The samples were washed twice, resuspended in 1.0 g CyFlow Space flow cytometer (Partec, Münster, Germany). The mean fluorescence intensity (MFI) of Rho123 was assessed on fluorescence channel 1. At least 10,000 events were assayed for each sample.

#### 2.9.3. Effects of Compounds on Microtubule

The effect of the identified compound on interphase on microtubules, mitotic spindles, nuclear morphology, and cell cycle was assessed. In order to do so, human lung carcinoma A549 cells were incubated for 20 h with serial concentrations of the diterpene ranging from 10 nM to 200 μM.

The DNA content was analyzed by flow cytometry using a BD FACSVerseTM cytometer (BD Biosciences, San Jose, CA, USA). Cells were collected and centrifuged at 500× *g*, washed with phosphate buffer saline (PBS), and resuspended in 50 μL of PBS. Following dropwise addition of 1 mL of ice-cold 75% ethanol, fixed cells were stored at −20 °C for 1 h. Samples were then centrifuged at 500× *g* and washed with PBS before resuspension in 1 mL of PBS containing 50 μg/mL propidium iodide and 100 μg/mL RNase A, and incubation for 1 h at 37 °C in the dark. The percentage of cells with decreased DNA staining, composed of apoptotic cells resulting from either fragmentation or decreased chromatin, was counted (minimum of 10,000 cells per experimental condition). Cell debris was excluded from analysis by selective gating based on anterior and right angle scattering.

## 3. Results and Discussion

The acetonic, ethanolic, and/or methanolic extracts of several *Plectranthus* species (Appendix A) were evaluated concerning their antimicrobial activity through the well diffusion method and further MIC and MBC determination. Furthermore, in order to evaluate their cytotoxic properties, toxicity assays were performed on *Artemia salina* L. and different cancer cell lines.

All 31 extracts were screened at a fixed concentration (1 mg/mL). Overall, five extracts demonstrated inhibitory activities against either or both *Enterococcus faecalis* (ATCC 29212) and *Staphylococcus aureus* (ATCC 25923) bacterial strains (Appendix A). All other extracts did not display antibacterial effects, showing inhibition zones lower than 10 mm (data not shown). The extracts prepared with acetone revealed better antimicrobial activity, particularly against Gram-positive bacteria.

A significant inhibition zone was recorded for the acetonic extracts of *P. aliciae* and *P. madagascariensis var. “Lynne”*, for both Gram-positive bacteria used (*E.faecalis* and *S. aureus*), with results similar to those of positive control. The remaining active extracts, however, showed selectivity only against *S. aureus*.

Regarding the results obtained on the well diffusion test, only Gram-positive bacteria were taken into account on these assays. Thus, in order to assess the lowest growth inhibitory concentration of the most promising *Plectranthus* spp. extracts, minimum inhibitory concentration values (MIC) were determined. Their bacteriostatic and bactericidal properties were also evaluated (Table 1) MIC values ranged from 15.6 to 125 μg/mL, while MBC values ranged from >62.5 to 250 μg/mL.

Comparing the MBC and MIC values of the selected extracts, and having in mind that a bactericidal effect is associated to a MBC no more than four times the MIC value [21], it is possible to conclude that the extracts are mainly bactericidal rather than bacteriostatic.

The preliminary evaluation of toxicity on *Artemia salina* is a fairly easy and inexpensive method and is particularly interesting when testing plant extracts as it can be used as a prescreen of their potential antitumor activities This assay also helps in gaining a preliminary understanding of which extracts may deserve additional testing. All 31 extracts were tested (data not shown), but only five of them exhibited toxic effects with values ranging from 13.62 to 110.76 μg/mL (Table 2). Curiously, these five acetonic extracts were the same that induced *S. aureus* growth inhibition. The lowest LC_50_, and consequently, the highest toxic potential, was attributed, once again, to the activity of *P. strigosus.* On the contrary, the least toxic extract was attributed to *P. stylesii*.

The GI_50_ values were also determined for each of the 31 extracts, in HCT116, MCF-7, and NCI-H460 cell lines, and the observed reduction in the cell growth was compared to the growth of cells treated with DMSO only (set as 100% growth). The results are summarized in Figure 1 below. The data bars not shown refer to a GI_50_ value equal or higher than 50 μg/mL.

According to the results, acetonic extracts revealed better cytotoxic effects. Five of these—*P. aliciae*, *P. japonicus*, *P. madagascariensis* var. *“Lynne”*, *P. stylesii*, and *P. strigosus*—have consistently exhibited a broad cytotoxic potential. When analyzing GI_50_ on the three tested cell lines, values ranged between 3.47 to 16.67 μg/mL, showing effective growth inhibition against specific cell lines.

*P. strigosus* acetonic extract displayed the highest overall anticancer potential, given its growth inhibitory activities towards HCT116 (GI_50_ = 3.78 ± 0.49 μg/mL), MCF-7 (GI_50_ = 8.35 ± 0.57μg/mL), and NCI-H460 (GI_50_ = 8.75 ± 0.70 μg/mL). On the other hand, the highest inhibitory growth activities on MCF-7 and NCI-H460 cell lines were attributed to the effect of *P. madagascariensis* var. “Lynne” (GI_50_ = 3.47 ± 0.15μg/mL and 5.39 ± 0.48, respectively).

The extract with the lowest anticancer potential was *P. reflexus* methanolic extract, for which GI_50_ was not reached in the screened range in all tested cell lines (GI_50_ ≥ 50 μg/mL).

Overall, *Plectranthus* spp. extracts recorded a broader cytotoxic activity against HCT116, given that 27 out of the 31 extracts tested have GI_50_ lower than 50 μg/mL.

According to the results obtained in the previous screening (Figure 1), the most promising *Plectranthus* spp. extracts were tested in MDR cell lines with P-gp overexpression: NCI-H460/R (non-small cell lung carcinoma cell line) and DLD1-TxR (colorectal adenocarcinoma cell line). The effects of the five selected acetonic extracts were assessed by SRB assay after 72 h, and their IC_50_ values were determined (Figure 2). *P. strigosus* acetonic extract exerted an overall best effect in both non-small cell lung carcinoma and colorectal adenocarcinoma cells, which showed IC_50_ values of 5.64 μg/mL (NCI-H460), 11.09 μg/mL (NCI-H460/R), 2.41 μg/mL (DLD1), and 2.51 μg/mL (DLD1-TxR).

All five extracts exerted similar results towards colorectal adenocarcinoma cells (DLD1 and DLD1-TxR). Thus, their efficacy might not depend on the presence of a MDR phenotype. Acetonic extracts from *P. aliciae, P. japonicas*, and *P. strigosus* were less active against non-small cell lung carcinoma MDR cells (NCI-H460/R) in comparison with their sensitive counterparts (NCI-H460). Interestingly, resistant NCI-H460/R were more sensitive to extracts from *P. malvinus* and *P. stylesii* showing collateral sensitivity, the phenomenon when MDR cells are more vulnerable than corresponding sensitive cells.

In order to assess the chemical composition of the most active extract (*P. strigosus* acetonic extract), and in an attempt to identify the compounds responsible for the referred activity, a combination of HPLC–DAD (diode array detector) –MS techniques were carried out. The HPLC–MS analysis showed two main peaks at 19.8 and 22.0 min, which revealed a UV spectrum characteristic of quinone methide diterpenes [29]. The ESI mass spectra of the compounds pointed to the presence of parvifloron E or F (peak at 19.8 min, *m*/*z* 451 [M + H]^+^) and parvifloron D (peak at 22 min, *m*/*z* 435 [M + H]^+^). The presence of parvifloron D was confirmed through MS/MS analysis by comparison with a standard compound. MS/MS spectra showed product ions at *m*/*z* 297 (base peak) and *m*/*z* 279, corresponding to the loss of the ester moiety ([M-C_7_H_5_O_3_ + H]^+^) and ester moiety and H_2_O ([M-C_7_H_5_O_3_-H_2_O + H]^+^), see Appendix A. The 19.8 min compound displayed a similar fragmentation pattern corresponding to the same diterpenic scaffold, with one more hydroxyl group on the aromatic ring substituent. Therefore, this compound was tentatively identified as parvifloron F. The quantification of parvifloron D was performed through HPLC–DAD/UV and was found to be present in the acetonic extract of *P. strigosus* with a relative concentration of 9%.

Given its antitumor properties [30], the presence of the latter may explain, to a certain extent, the foreseen toxic properties of *P. strigosus* acetonic extract. In order to further explore its cytotoxicity, additional studies were performed on this compound.

Afterwards, the preliminary toxicity evaluation of parvifloron D on *Artemia salina* L. resulted in a 76.6% mortality rate on this species after 24 h incubation.

In addition, its cytotoxic activity was further evaluated on NCI-H460 and NCI-H460/R cell lines. To determine whether parvifloron D is selective towards cancer cells, the cytotoxic effect in normal embryonic bronchial fibroblasts (MRC-5) was also evaluated.

As expected, this diterpene demonstrated high cytoxicity and exerted a similar activity in all three cell lines, with IC_50_ ranging from 1.7 to 1.9 μM. Interestingly, it also showed the same efficacy in sensitive and MDR cancer cells, which implies that parvifloron D is able to evade P-gp efflux activity (Figure 3A). However, it was equally active against normal cells (Figure 3A). Parvifloron D was also tested regarding its potential to inhibit P-gp activity in NCI-H460/R cells. Unfortunately, parvifloron D was not able to increase the Rhodamine 123 accumulation in P-gp overexpressing cells (Figure 3B). Considering its high cytotoxic potential, further examinations of parvifloron D should be performed. Its mechanism in cancer and normal cells, with respect to the application of lower concentrations and different time scheduling, are necessary to clarify whether this compound should be developed as an anticancer agent.

In order to check whether this abietane diterpene was a microtubule stabilizer agent, A549 cell morphologies in the IC_50_ culture plates were examined with an inverted microscope, after 20 h incubation with parvifloron D. Paclitaxel was used as control tubulin specific drug. All microtubule targeting drugs, stabilizers, or depolymerizers in contact with A549 cell lines induced the appearance of round cells (mitotic cells) that detach from the plate. However, parvifloron D only displayed a cytotoxic effect rather than cytostatic, with no mitotic blockage, practically ruling out an action on mitotic spindle microtubules (data not shown). These results suggest that the antibacterial and anticancer activities of the acetonic extract of *P. strigosus* are attributed to the presence of parvifloron D. These results correlate to the representative HPLC–DAD/UV (λ = 254 nm) chromatogram of acetonic extract of *P. strigosus* and UV DAD spectra of peaks. The prepared acetonic extract of *P. strigosus* (c = 0.67 mg/mL) was revealed to have Parvifloron D (9% (*w*/*w*) in its composition.

## 4. Conclusions

Overall, the screening of the *Plectranthus* spp. through several extracts carried out in this work was motivated by the possibility of enlightening the cytotoxic activities of these species. Acetone extracts have been proven to possess higher overall antimicrobial and cytotoxic activity. Comparing the results of the well diffusion method with the corresponding GI_50_ (μg/mL) values obtained in cancer cells, it can be concluded that the extracts with antimicrobial properties in *S. aureus* were also those with the lowest GI_50_ values. Therefore, among the 31 *Plectranthus* spp. extracts, the acetonic extracts from *P. aliciae*, *P. japonicus*, *P. malvinus* van Jaarsv. and T.J. Edwards, *P. strigosus*, and *P. stylesii* were selected as promising candidates for further examination. To the best of our knowledge, this study provides the first insight on the biological activities of some *Plectranthus* spp. such as *P. aliciae*, *P. madagascariensis* var. “Lynne”, and *P. stylesii*. More importantly, it has also allowed the assignment of components that might be responsible for their activity, such as the abietane diterpene parvifloron D. Overall, the acetonic extract of *P. strigosus* revealed the best cytotoxic potential.

For the reasons mentioned above, the results demonstrated that the *Plectranthus* genus may contain therapeutically useful compounds against both Gram-positive bacteria and some solid cancer cell lines including those with MDR phenotype.

## Figures and Tables

**Figure 1 biomolecules-09-00616-f001:**
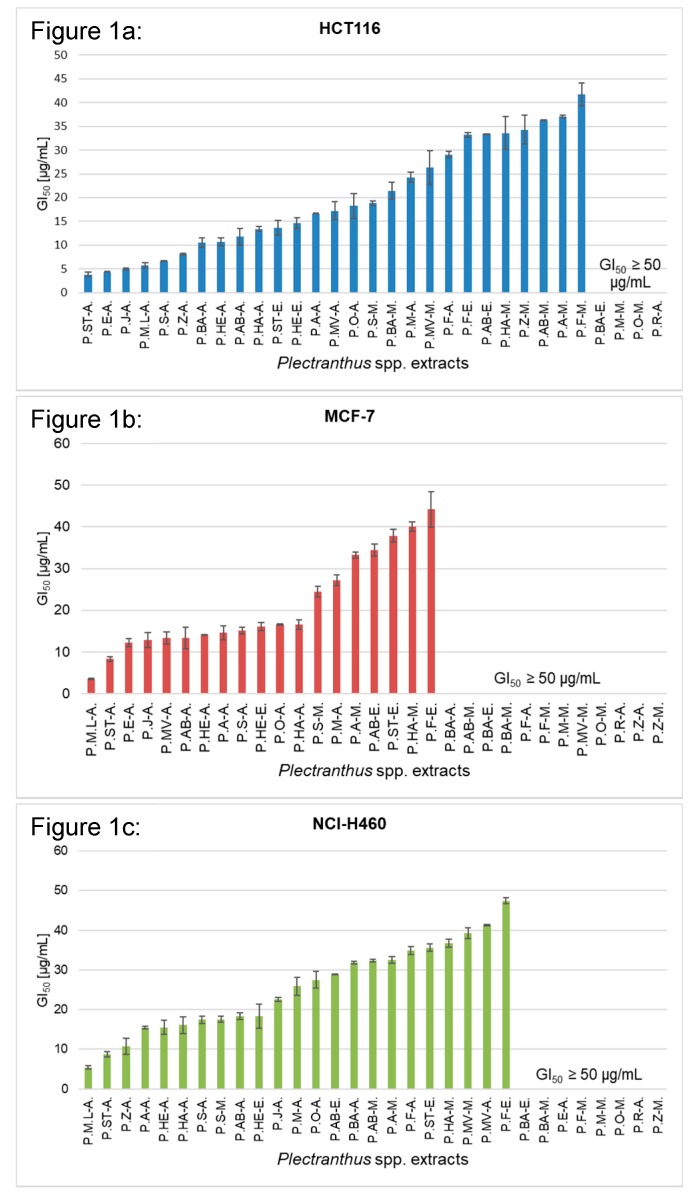
Growth inhibition of 50% (GI_50_) values (μg/mL) of *Plectranthus* spp. extracts in HCT116 (**a**), MCF-7 (**b**), and NCI-H460 (**c**) cells were determined after 48 h treatment using the SRB assay. Data are mean ± SEM (*n* = 3,4). Doxorubicin was used as a positive control (GI_50_: 54.00 ± 3.24 ng/mL in HCT116, 85.17 ± 4.10 ng/mL in MCF-7, 292.01 ± 2.32 ng/mL in NCI-H460 cells).

**Figure 2 biomolecules-09-00616-f002:**
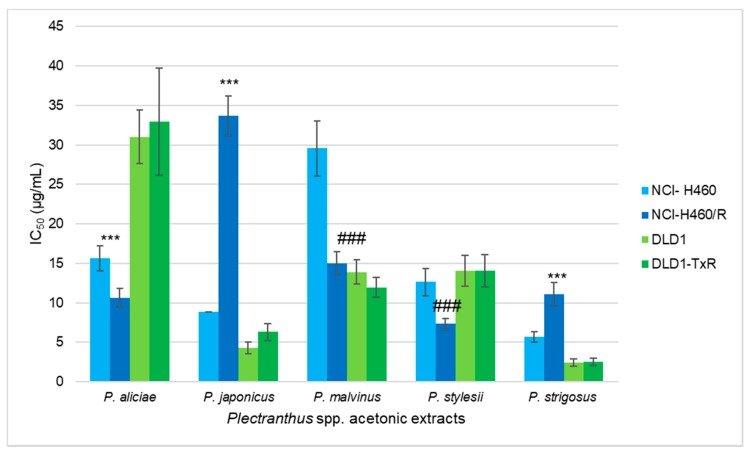
Sensitivity of MDR (MultiDrug Resistance) cancer cells and their corresponding sensitive cells to acetonic *Plectranthus* extracts. The IC_50_ (half maximal inhibitory concentration) was determined using SRB assay after 72 h treatment with acetonic extracts. Statistical analysis was performed in GraphPad Prism 6 by two-way ANOVA: *** indicates the resistance to the specific extract (*p* < 0.001); ### indicates the collateral sensitivity to the specific extract (*p* < 0.001).

**Figure 3 biomolecules-09-00616-f003:**
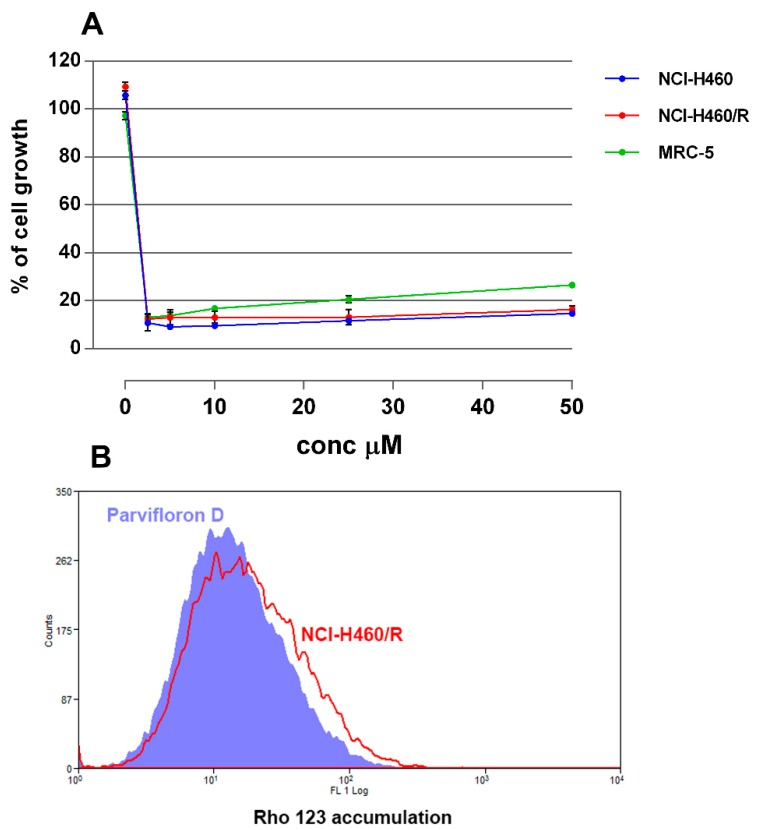
Anticancer effects of parvifloron D. The cytotoxicity was assessed by MTT—3-(4,5-Dimethylthiazol-2-yl)-2,5-diphenyltetrazolium bromide—assay after 72 h treatment in NCI-H460, NCI-H460/R, and MRC-5 cells (**A**). Rho 123 accumulation was evaluated as an indicator of P-glycoprotein (P-gp) function in NCI-H460/R cells untreated and treated with parvifloron D (**B**).

**Table 1 biomolecules-09-00616-t001:** Minimum inhibitory concentration (MIC) and minimum bactericidal concentration (MBC) values of *Plectranthus* spp. acetonic extracts against tested bacteria (results expressed in μg/mL).

Acetonic Extracts	Gram-Positive Bacteria			
	*E. faecalis* (ATCC 29212)		*S. aureus* (ATCC 25923)	
	MIC	MBC	MIC	MBC
*P. aliciae*	15.6	125	62.5	125
*P. japonicus*	62.5	125	62.5	125
*P. madagascariensis* ‘Lynne’	15.6	>125	125	125
*P. strigosus*	15.6	>62.5	125	125
*P. stylesii*	31.3	>125	62.5	250
Positive control (VAN)	1.95		7.82	
Negative control (DMSO)	>500		>500	

VAN—vancomycin (1 mg/mL); NOR—norfloxacin (1 mg/mL).

**Table 2 biomolecules-09-00616-t002:** *A. salina* general toxicity results LC_50_ (μg/mL) of acetonic extracts of the selected *Plectranthus* spp.

Acetonic Extracts	Mortality Rate (%) at 100 ppm Concentration	LC_50_ (μg/mL)
*P. aliciae*	16.71 ± 1.01	53.48
*P. japonicus*	21.85 ± 0.35	38.9
*P. madagascariensis* var *“Lynne”*	9.67 ± 0.93	91.7
*P. stylesii*	16.30 ± 0.99	110.76
*P. strigosus*	42.88 ± 2. 21	13.62
DMSO	22.50 ± 3.54	N.A.

N.A.—non applicable.

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
