# Peer review of "Parvifloron D from Plectranthus strigosus: Cytotoxicity Screening of Plectranthus spp. Extracts"

_biomolecules, 2019, doi:10.3390/biom9100616_

Round 1

Reviewer 1 Report

Lines 93, 94: the name „Saccharomyces cerevivisiae” should be corrected. It should be „Saccharomyces cerevisiae”.

Lines 96, 108: Why was the 37 °C incubation temperature used for yeast? Under these conditions, yeast does not grow or very poorly. Growth temperature of S. cerevisiae ATCC 9763 is 30°C (see description of growth conditions in: 9763.pdf). Typical S. cerevisiae colonies are observed at 25°C.

Lines 99, 105: The plural of “inoculum” is inocula, not “inoculums”.

Lines 126-130: It should be written: Thus, a swab from each well where no growth was observed - containing the tested extract concentration series - was plated on a Mueller Hinton Agar. The Petri plates were then incubated for 24h at 37ºC, for further evaluation.  The lowest concentration that revealed no visible bacterial growth after sub-culturing was taken as its MBC.

Author Response

Dear Reviewer 1:

We appreciate the reviewer 1 comments, which have helped us to improve the manuscript. We have carefully considered the suggestions, addressing and incorporating them in the manuscript as detailed below. The modifications in the corrected paper have a yellow background.

Reviewer #1: Detailed comments:

Comment 1 Lines 93, 94: the name „Saccharomyces cerevivisiae” should be corrected. It should be „Saccharomyces cerevisiae”.

Authors answer: We thank the reviewer for the correction. All the corrections were incorporated in the text as suggested.

Comment 2 Lines 96, 108: Why was the 37 °C incubation temperature used for yeast? Under these conditions, yeast does not grow or very poorly. Growth temperature of S. cerevisiae ATCC 9763 is 30°C (see description of growth conditions in: 9763.pdf). Typical S. cerevisiae colonies are observed at 25°C.

Authors answer: We thank the reviewer for the correction. In deed the growth temperature is 30 C, so we change the manuscript and incorporated in the text as suggested.

Line 96: “Cultures of microorganism were grown at 37 ºC (30 ºC for the yeasts) for 24h and were maintained on respective agar slants at 4 ºC.”

Line 108-109: “Plates were then incubated at 37 °C (30 ºC for the yeasts) for 24 h.”

Comment 3 Lines 99, 105: The plural of “inoculum” is inocula, not “inoculums”.

Authors answer: We thank the reviewer for the correction. The corrections were incorporated in the text as suggested.

Comment 4 Lines 126-130: It should be written: Thus, a swab from each well where no growth was observed - containing the tested extract concentration series - was plated on a Mueller Hinton Agar. The Petri plates were then incubated for 24h at 37ºC, for further evaluation.  The lowest concentration that revealed no visible bacterial growth after sub-culturing was taken as its MBC.

Authors answer: We thank the reviewer for the correction. The corrections were incorporated in the text as suggested.

Reviewer 2 Report

The manuscript "Parvifloron D from Plectranthus strigosus: cytotoxicity screening of Plectranthus spp. extracts" has been significantly improved and the authors have taken into consideration the majority of the concern given in previous round. Manuscript could be published but its presentation needs an improvement.

Some concerns, mostly of them regarding the quality of presentation, are:

-Introduction is quite small and only contains 10 references. It could be enlarged to provide a better overview of the field.

-Figure 1. Size of the titles of the subfigures is still quite big, even more than the titles of the sections of the manuscript, it should be reduced and centered.

-Figure 1. I notice that the extracts are in descendant order of activity. For not being confusing to the readers, it would be better that the order would be the same in the three subfigures.

-Figure 2. Adding the title of the figure is not necessary as all the information is at the figure caption. Besides, it applies the same comment than in figure 1: the size of the letters of both the title and axis is too big. If a smaller size had been used, all the x-axis caption would have entered in the same line.

-Conclusion: its beginning has no sense. "The acetonic Overall,..." Is anything missed?

Author Response

Dear Reviewer 2:

We appreciate the reviewer 1 comments, which have helped us to improve the manuscript. We have carefully considered the suggestions, addressing and incorporating them in the manuscript as detailed below. The modifications in the corrected paper have a yellow background.

Reviewer #2: The manuscript "Parvifloron D from Plectranthus strigosus: cytotoxicity screening of Plectranthus spp. extracts" has been significantly improved and the authors have taken into consideration the majority of the concern given in previous round. Manuscript could be published but its presentation needs an improvement. Some concerns, mostly of them regarding the quality of presentation, are:

Detailed comments:

Comment 1 Introduction is quite small and only contains 10 references. It could be enlarged to provide a better overview of the field.

Authors answer: We thank the reviewer for the suggestion. We include more information in the introduction:

“Diterpenes are frequently found in the Plectranthus genus, and are compounds of high interest, considering their wide spectrum of biological activity, characterized by antimicrobial and anti-inflamatory activity, but also, anticancer properties11.

The majority of diterpene metabolites isolated from Plectranthus spp. belong to abietane, kaurane, and labdane classes11. The more prominent functional groups are phenol or quinone diterpenes belonging to royleanone, coleon or parvifloron abietanes, which has shown to be potent anticancer agents classes12,13. It is known that the major component of the essential oil of P. madagascariensis is the diterpene 6,7 dehydroroyleanone, which is easily identified due to its characteristic orange reddish color14;15  and it has already shown cytotoxic properties16,,17. On the other hand, it is also described antitumoral activity by a number of other Plectranthus species such as P. amboinicus, P. barbatus and P. hadiensis 2, so the screening of their extracts is a significant achievement. Other previous studies showed that natural compounds with antitumor activity, isolated from Plectranthus, induce apoptosis by caspases activation. The discovery of natural tumor-promoting phorbol esters acting as Protein Kinase C (PKC) activators, lead to a new interest in the role of these proteins, but also, and more importantly, to new studies focusing on diterpenes as PKC modulators. It is known that some of the PKC isoforms are implied in tumoral regression while others are involved in tumoral invasion, so modulating its activity is an important issue18.”

Comment 2 Figure 1. Size of the titles of the subfigures is still quite big, even more than the titles of the sections of the manuscript, it should be reduced and centered.

Authors answer: We thank the reviewer for the suggestion. We change the Figure 1 accordingly.

Comment 3 Figure 1. I notice that the extracts are in descendant order of activity. For not being confusing to the readers, it would be better that the order would be the same in the three subfigures.

Authors answer: We thank the reviewer for the suggestion. We change the Figure 1 accordingly.

Comment 4 Figure 2. Adding the title of the figure is not necessary as all the information is at the figure caption. Besides, it applies the same comment than in figure 1: the size of the letters of both the title and axis is too big. If a smaller size had been used, all the x-axis caption would have entered in the same line.

Authors answer: We thank the reviewer for the suggestion. We change the Figure 1 accordingly.

Comment 5 Conclusion: its beginning has no sense. "The acetonic Overall, ..." Is anything missed?

Authors answer: We thank the reviewer for the correction. The beginning is now starting with: “Overall, the screening of the Plectranthus spp. through several extracts carried out in this work was motivated ….”

This manuscript is a resubmission of an earlier submission. The following is a list of the peer review reports and author responses from that submission.

Round 1

Reviewer 1 Report

The manuscript is very interesting and provides a lot of information about the type of Plectranthus and its possible actions against both Gram-positive bacteria and some solid cancer cell lines. However, there are significant shortcomings in the "methodology" and "Results" sections, which the reviewer cannot ignore in his assessment. Generally, this concerns the lack of results of the antifungal activity and errors in the methodology of MBC. For this reason, I believe that the authors should complete the results that are not available in this work and submit the manuscript for evaluation again.

Detailed comments:

Lines 2-3: the Latin names should be in italics. "Extracts" written with a lowercase.

Line 90: Provide the reason for the test with the Saccharomyces cerevisiae strain. S. cerevisiae is classified as a level 1 biosafety and is not a pathogenic genus, unlike the other selected yeast species, C. albicans.

Lines 92-93: under which conditions the yeast strains were grown. Please complete the description of the method.

Line 96: What does a 1 mg / mL concentration mean? Is 1 mg dry mass of the extract in 1 ml, or 1 mg liquid extract was 1 ml of solution?

Line 94-107: Were only the bacteria examined by the well diffusion method, and the yeast not? Haven’t the extracts been tested against yeast strains?

Line 110: there are no results related to the yeast strains, so it should be "the antibacterial activity" and not "the antimicrobial activity".

Line 115: what was the initial number of bacterial suspensions in 10 μl of the suspension? Was it equivalent to 0.5 ° Mc Farland’ density? Please, specify?

Lines 119-124: The method is incorrect. If Mueller Hinton Agar has previously inoculated with the correspondent bacteria, then the MBC cannot be determined. The culture is transferred from the wells to the uninoculated MHA and counts the grown colonies after incubation.

Lines 261-266: The authors discuss the antibacterial activity of extracts and do not discuss antheld activity. Why? The lack of these results is a serious drawback of the work being assessed.

Lines 276-278, Table 1: The results of MIC and MBC / MFC of the six test strains given in the methodology (lines: 88-90) should be presented in the table: Enterococcus faecalis (ATCC 29212), Escherichia coli (ATCC 25922), Pseudomonas aeruginosa (ATCC 27853), Staphylococcus aureus (ATCC 25923), and fungal strains Candida albicans (ATCC 10231), Saccharomyces cerevivisiae (ATCC 9763). The lack of these results prevents the reader from actually evaluating the antimicrobial activity of the extracts.

Reviewer 2 Report

The Plectranthus genus is used as traditional medicine to treat illnesses. In this study, the authors prepared 31 extracts from 16 Plectranthus species and evaluated their respective antibacterial and anticancer activities. The MIC, MBC, and cytotoxicity data showed that five acetonic extracts  possessed potential activities. With the help of HPLC-MS and MS/MS analysis, the authors suggested that the bioactive compound could be parvifloron D, a cytotoxic diterpene molecule. Overall, this manuscript was well designed and the methods were clearly described. Hence, I recommend this manuscript to be published on Biomolecules after minor revision.

Minor issues:

1. Several sentences are grammatically wrong, please carefully check the whole manuscript.

2. There are no Figure 2 and 3.

3. The authors suggested the antibacterial and anticancer activities of the acetonic extracts were attributed to the presence of parvifloron D, which seems inadequate to make the conclusion. I would be more convincing if they quantitatively correlate the amount of parvifloron D in the extracts with the observed bioactivities.

4. Page 8 line 295 "equal or higher than 50 @g/mL" should be corrected as "equal or higher than 50 μg/mL"

5. In table 2. The mortality rate of DMSO at 100 ppm is higher than the tested samples except P. strigosus,  is this data correct?

Reviewer 3 Report

The manuscript "Parvifloron D from P. strigosus: cytotoxicity screening of Plectranthus spp. extracts" is interesting, provides significant novel findings and is well written. However, I would encourage authors to fix a few minor concerns before publication, mostly of them with the intention of improving the quality of presentation:

-For making the article more attractive at first sight (it will be reflected in more citations) and to help those readers that are not experts in botanical sciences I would recommend to include (at least in Supplementary) images of selected plants among the ones studied. I am conscious that including one per each different plant can be a problem and would provide an excess of information; but including a few representative ones (2-5) could be nice.

-I would encourage authors to revise the format of the figures. Size of the figure title is too big, and more taking into account that the same information (and expanded) is in the legend of the figure. I would reduce the size of the caption at least to a size similar to the one of the text.

-As minor point, in same places the unit is in a different line than the number value. For example, 10 mm in lines 111-112, 25 mg/ml in lines 141-142. 1 ml in lines 249-250. Please put the two in the same line using a hard space (Control-Shift-Space) instead a normal space.

-It is advisable that tables are not divided in two pages, as happens in pages 7 and 8 with Table 2. It could be achieved easily reduced the interlined space in Table 1, which is quite high.

-Legend of Figure 1 is not consistent with the text (cited as Figures 1-3), please homogeneize (and if necessary, renumber Figures 4-5). I would recommend dividing in three figures, so that 2 appear in Page 8 and 1 in page 9, or reducing slightly the size for not divide it between 2 pages. Besides, the abbreviations of the the Plectranthus species in this figure would be better in italics.

-If available, it would be nice if authors provide the MS/MS spectra commented and interpretated in Page 10, to illustrate better the paragraph.

-Finally, manuscript would be more attractive to readers if the last two paragraphs of the manuscrip (lines 379-393) are provided as conclusions.